# Tribological Characterization of Micro Ball Bearings with and without Solid-State Lubrication

**DOI:** 10.3390/mi14091775

**Published:** 2023-09-16

**Authors:** Mateusz Jakub Samsel, Artur Chańko, Marcin Michałowski, Miguel Fernandez-Munoz, Efren Diez-Jimenez

**Affiliations:** 1Faculty of Physics, Warsaw University of Technology, 00-662 Warszawa, Poland; mateusz.samsel.dokt@pw.edu.pl; 2Faculty of Mechatronics, Warsaw University of Technology, 02-525 Warszawa, Poland; artur.chanko.stud@pw.edu.pl; 3Mechanical Engineering Area, Universidad de Alcalá, 28-805 Alcalá de Henares, Spain; miguel.fm@uah.es (M.F.-M.); efren.diez@uah.es (E.D.-J.)

**Keywords:** micro-ball bearing, tribology, friction, MEMS, solid lubrication

## Abstract

The tribological characteristics of a below 1 mm micro ball bearing comprising steel disc and cages coated with thin copper and silver films were investigated. Electroplating and laser cutting were used to manufacture used elements. Friction was measured using a linear stage and an adapted version of a friction-loop method. The obtained results show an interesting relationship between the geometric properties of the micro scale thrust bearing and their performance and operational stability, which can be correlated to similar relationships observed in the macro scale. The most optimal design of the bearing showed stable operation, with the simplified rolling resistance coefficient in the range 0.002 to 0.003, independently of applied load, which was in range 150 mN to 1500 mN. The possibility of creating easily manufacturable micro ball bearings with a low rolling resistance coefficient comprised solely of cheap and sturdy elements was shown.

## 1. Introduction

Recent years have shown an increasingly progressive need to miniaturize electromechanical devices. Electromechanical devices often need components that facilitate the mutual movement of elements for proper operation. Small linear motions are common in MEMS (microelectromechanical system) applications [1]. However the sliding operation is more commonly used in micromechanics compared to applications using ball bearings, such as in the watch industry; this leads to higher friction and wear. The micro ball bearings were already successfully used in pumps [2], power generators [3], and rotary microactuators [4]. However, current versions of the micro ball bearing designs are purely based on silicon [5]. This leads to complicated manufacturing processes, poor mechanical performance, and the short lifespan of the mechanism [6,7]. On the macro scale, these problems could be facilitated using carefully selected lubrication, but lubricants are often incompatible with MEMS because of the precision requirements of devices [8]. The friction and wear can be influenced using solid lubricants, which can be also used as protective coating, which could also work as protection from the corrosive environment [8]. Numerous simulation models were previously developed considering the sliding operation [9,10]. The motivation behind this research was to explore the possibility of transferring common aspects found in macro ball bearings to the design of micro ball bearings, thereby creating a platform for designing reliable linear and rotary devices with sizes below 1 mm for micro-scale applications using simple, cheap materials, and manufacturing processes composed of only few easily replicable steps. To achieve this goal, this work explores the design and testing of novel type of linear micro ball bearings with the sizes below 1 mm. While silicon-based micro ball bearings were previously studied, there is still a gap in the research of steel-based micro ball bearings, especially with the use of a cage. Steel-based designs allow for the use of laser-cutting as a fabrication technology, which can provide easy access to both mass production and prototype testing. During the testing, the rolling resistance coefficient was measured to assess the efficiency and lifespan of the proposed device.

## 2. Device Design

### 2.1. Proposed Device

The proposed bearing is composed of two steel plates, one steel cage, and seven soda lime glass microspheres. The austenitic non-magnetic stainless steel was selected as a main construction material because it is a well known and widely used material and it shows great compatibility with commonly used solid lubricants [11]. The commercial availability of thin metal sheets made of austenitic steel is another benefit. The cage was used to avoid the jamming of the rolling elements and to introduce stability in the bearing performance. Additionally, the cage limits the collisions between micro balls during the operation. The collisions between rolling elements were proven to increase rolling resistance coefficient even 1.5 times [12]. Multiple cage designs were tested using different heights of cages and different coating materials to select the most optimal solution. For the cage, the race riding design was selected; roller riding bearings are commonly used in the industrial applications, but they are unsuitable for self-lubricating bearings because wear on contact between the ball and cage leads to a very quick loss of guidance. Soda lime glass microspheres are chosen due to the commercial availability and precise dimensions of the manufactured elements. The usage of glass balls was proven to significantly reduce the friction and wear of the system [13]. Additionally, glass has high strength and low surface energy [14], which lowers adhesive forces in the contact area in normal conditions [15]. Spacer-grade 98–102 μm microspheres produced by Cospheric were used. According to the producer, their high sphericity, close size tolerances, and high mechanical strength and chemical durability make them ideal for all applications that require these high-grade standards. The render of the proposed device built using the above-mentioned elements is presented in Figure 1.

### 2.2. Fabrication

One of the goals of our research was to propose an easy design that can be easily replicated using standard machining methods. Except for micro ball assembly, all other processes are compatible with laser cutting technology and suitable for production at a large scale. The studied bearing is composed of three elements organised in stack layout: top and bottom planar discs and cage in-between. The planar discs with an outer diameter of 1 mm were cut out in multiple batches using LPKF StencilLaser G 6080 from 1 mm-thick stainless-steel sheet. Each batch contained hundreds of discs. The obtained discs were cleaned using ultrasonic cleaner with isopropyl alcohol (IPA) and deionized water. After cleaning sample discs were studied under a microscope and based on the obtained results, the decision was made to accept or to redo the batch. In the next step, the ball cages were cut out of the steel sheets with thicknesses, respectively, of 30, 40, 50, 60, and 70 μm. The laser parameters were selected based on trial and error for each of the thicknesses of steel. The outer diameter of the cages was set to 0.8 mm, and the ball housings’ diameter was nominally set to 130 μm. The cages are also cleaned using ultrasonic cleaner with IPA and deionized water. After cleaning, a batch of 40 μm steel cages was selected and coated with copper and silver using the electroplating process. During electroplating, the negative pole of the power supply was connected by wires to the cages, and the positive pole was connected to a platinum-coated electrode. We selected the process parameters to obtain a coating of 5 μm thickness over the entire surface of the separator. By method of trial and error, the nominal voltage of 2.5 V and a current limit of 50 mA for the silver coating and 100 mA for the copper coating were selected. The plating bath time was 45 and 65 s, respectively. The process was performed at 22 °C, and the solution was stirred with a magnetic stirrer at 200 rpm. Once the coating was completed, the cages were cleaned with a bath of deionized water and then dried. We selected a representative random sample (10 elements of each type) from hundreds of the prepared elements. The selected components were measured under optical microscope, and their surfaces were investigated (Figure 2a). The measurement of the surface profile was used to reject elements that had flow-outs (manifested by the high profile in the picture) and distortions (also visible on the profile) after laser cutting and electroplating and thus differed from the rest of the elements. Figure 2a) shows a correctly made separator without distortions. The obtained dimensions of the elements are presented in the Table 1, where $\bar{h}$—average thickness, $\bar{d_{in}}$—average inner hole diameter, $\bar{d_{out}}$—average outer diameter, and $\bar{d_{b}}$—average ball housings diameter. The measurements were carried out using a glass calibration plate with 1 μm precision. The example of obtained element fitted with soda-lime glass balls is shown in Figure 2b.

### 2.3. Assembly

The proposed assembly station is composed of a 3-axis high precision system providing a precision of movement up to 1.25 μm, and the maximum travel range is 30 mm. The devices collect optical feedback from the stereo microscope and side cameras. Component placement is carried out using negative pressure with a thin nozzle attached. The negative pressure is obtained with a PWM (pulse-width modulation)-controlled vacuum pump. The vacuum pump is connected to the Quant X Fisnar syringes, which are fitted with different gauge needles. The size of the used needle is selected based on the size of the element moved, assuming that the suction force is proportional to the cross section of the needle tip; in this case, G34 and G36 needles were used for cages and balls, respectively. The assembly station allows for a wide range of manipulation, starting with the simple movement of elements to make precise connections such as for gluing. The assembly and measurements were carried out in a strictly controlled environment with a temperature set to 20 °C and humidity set to 19% because humidity changes were proven to influence the micro ball bearings’ performance [16]. After the element is picked up, it can be safely moved in the XYZ space, manually using a joystick or automatically using scripts. To slowly drop the element, we lower the duty cycle of the PWM control signal. The device is presented in Figure 3. The system used is an adaptation of a similar stage used in the assembly of electronic components called SMT (surface mount technology).

The full assembly process of the proposed ball bearing is as follows:The preparation of necessary equipment. Needles and bigger elements are cleaned in the ultrasonic cleaner by first using acetone/IPA; afterwards, deionized water is used and they are dried. This prevents adhesion due to viscosity and contamination with organic material.The lower disc is picked up and placed on the leveled surface.The cage is picked up and placed on top of the lower disc.Holes are identified using the microscope and cameras. Balls are picked up and placed inside of the ball housings.The upper disc is placed on top of the setup.

## 3. Test Setup

To characterize the dynamic properties of the constructed bearings, we measured the simplified rolling resistance coefficient defined as:(1)$$\mu =\frac{T}{N}$$
where *T* is the resistance tangential force and *N* is the normal load applied to the system. We propose an experimental setup presented in picture Figure 4 to determine this rolling resistance coefficient.

To approximate the behavior of the final bearing, linear movement with three cages filled up with balls between two metal plates was used. All of the elements are kept stacked together to assure the even spread of electric charges. The bottom plate is later connected to the ground to eliminate any electrostatic forces. The centers of cages are placed evenly on a diameter of 8 mm circle using the previously described assembly station. Later, the metal plate is placed on top of the balls using the same station. This ensures the even spread of loads between balls and cages. To load the system, precise weights are used, and even spread of mass through the weight is assumed; the center of the weight is calculated based on the three-point measurement of the diameter of the weight from top view camera. Later, the center of the weight is aligned with the center of the top steel plate.

We use a system consisting of a PWM4M load cell, angle bracket, and XYZ motion stage to perform the measurement. The load cell allows for measurements with a resolution up to 1 mN, and the XYZ allows for precise positioning up to 1.25 μm. The angle bracket is loosely connected to the XYZ motion stage; this part of the mechanism acts as a rolling support. Thus, the connection allows for the transfer of horizontal load only. The angle bracket is placed on top of the steel plate; the width of the angle bracket is smaller than the diameter of the top steel plate to ensure the possibility of finding the center of the mass. The angle bracket and top steel plate work as an initial load applied to the system, which increases the stability of the system before the measurement starts. During each measurement, multiple 1 mm passes are made in both directions under one constant load. The axial load is monitored using a load cell. The obtained signal is collected and saved. In the next step, we fit the step function to the obtained signal to approximate the ideal behavior of the bearing without any operational noise. The obtained two directional signals are then subtracted from each other. In this way, we obtained the so-called friction-loop characterizing the friction in the system. An example of this measured signal is shown in Figure 5. This approach to friction force calculation is commonly used during the friction measurements in atomic force microscopy [17] (AFM) because it removes misalignment bias [18].

## 4. Results and Discussion

### 4.1. Normal Load Impact

In the first step, the relation between rolling resistance coefficient and normal load was investigated for different cage thicknesses. The obtained results can be seen in Figure 6.

The obtained points do not support any significant relationship between load and rolling resistance coefficient. This is contrary to the research performed by Hanrahan et al. [16], who described how the friction in the micro scale bearings should be mostly influenced by the adhesive forces due to the effects of scale. This would lead to the frictional force scaling with applied load *N* as $N^{\frac{2}{3}}$. However, during our research, we paid special attention to limit the impact of humidity as much as possible, and we selected materials displaying very low adhesion between each other. Thus, we conclude that the rolling resistance coefficient in our study is only a function of the materials used and the bearing geometry, assuming that the simplified rolling resistance coefficient can be used to properly characterize the dynamic properties of the studied system.

### 4.2. Impact of Cage Race Clearance

The cage clearances influence bearing stability wear and friction; the impact of the cage clearance from the theoretical and experimental perspective in the macro scale was widely studied by Gupta et al. [19,20,21]. However, as far as we know, no quantitative and qualitative study of the impact of the bearing clearances was performed in the micro scale. During our research, we focused on the cage-race clearance, otherwise called the guiding clearance; the ball housings clearance was nominally set to 30 μm. The guiding clearance can be defined as C=d−h, where *d* is the nominal ball diameter and h is the cage height. For each height, about 900 data points were collected during multiple measurements with a constant speed set to 100 μm/s. Each measurement was performed using a new set of balls and separators to check repeatability and to validate previously observed behaviour. The obtained points are represented as boxplots in Figure 7.

To properly analyze the obtained results, ANOVA in form of the Kruskal–Wallis H-test [23] was performed with the following hypothesis: H0 there is no difference between groups; H1: there is difference between groups caused by thickness. The obtained *p*-values indicate a difference in the rolling resistance coefficient depending on the cage’s construction. For the 30 μm thickness, the observed values do not exceed $\mu_{0.99}=0.009$ with 99% probability with the average rolling resistance coefficient $\bar{\mu }=0.003$; similarly, for the 40 μm cage the observed values do not exceed $\mu_{0.99}=0.008$ with the $\bar{\mu }=0.002$. For the 50 μm, $\mu_{0.99}=0.020$ with the $\bar{\mu }=0.006$; 60 μm
$\mu_{0.99}=0.021$ with the $\bar{\mu }=0.006$; and for the 70 μm, $\mu_{0.99}=0.022$ with the $\bar{\mu }=0.007$. These results indicate that the choice of the 40 μm thickness would be most optimal. We assume that the obtained results are closely related to the problem of cage instability. We have defined three scenarios; one of theses is when the height of the cage *h* is greater than the ball radius *r* the contacts between the balls and the cage can be modelled using the Hertzian model for contact between ball and plane; in this scenario, the height of the cage increases the likelihood of random contacts between the cage and the upper raceway, which can lead to self-locking and an increase in average friction and wear at the cage raceway contacts. The self-locking also substantially decreases the stability of the bearing and leads to the high observed frictional forces and high observed variance. The second scenario is when h<r; in this scenario, there is no self-locking, but the balls are trying to roll over the cage, which leads to the change in the direction of the reaction forces, which now oppose the rolling motion of balls. This increases overall friction in the system. The fact that the balls are trying to roll over the cage leads to increased wear on the ball-cage contact. The last and the most optimal scenario is when h=r; in this scenario, the likelihood of self-locking is minimal, there is no reaction force opposing rolling motion, and there are no conditions facilitating increased wear on the element’s interfaces. This scenario is realized for the cages with a height close to 40 μm due to the uncertainties surrounding cage height and ball diameter. Similar conclusions were drawn in the macro scale by Wen et al. [24].

### 4.3. Cage Coating Impact

Additionally, the influence of the solid lubricants on the device friction coefficient was investigated. Approximately 200 data points were collected for each of the coating materials. Each of the points was collected under a constant load of 40 g. The observed rolling resistance coefficient ranges match the results obtained during the investigation of thickness impact. The obtained data are present in Figure 8. Afterwards, ANOVA analysis was performed. Due to the coating process, the geometric dimensions of the coated 40 μm cages grew closer to the observed dimensions of the 50 μm cages. No significant impact of the coating on the frictional forces in the system was observed if we consider the changes in the cage heights introduced by the coating process.

### 4.4. Prolonged Operation Impact

To assess the impact of the coating and cage thickness on the bearing operational stability and wear, we performed 8 h long measurements during which the bearing elements passed the distance of approximately 3 m, which corresponds to 1000 full back and forth cycles (Figure 9). The measurements were performed under a constant load of 40 g. During this measurement, no significant changes in the friction coefficient were observed for the bearings with and without coating with the cage height less than 60 μm. For the thicker cage, fluctuations of the rolling resistance coefficient were observed, which can be explained by the operational instability introduced by the small clearances. After each measurement, the bearing balls and raceways were studied under the microscope, but no measurable wear was seen. Additionally, one 72 h, which corresponds to 9000 cycles, measurement was performed for the 40 μm bearing, but again no noticeable wear was observed.

### 4.5. Comparison with Literature

In recent years, numerous studies have investigated the friction behaviour of micro ball bearings, with the aim of improving their efficiency and durability. To evaluate the validity of our experimental results, it is important to compare them with the results obtained by other researchers in the field. Dae-san et al. [25] studied the performance of Si micro ball bearings working within v-shaped grooves coated with the ultra-thin protective layers containing borosilicate glass balls with a diameter under 60 μm. The measurement conditions were very similar to the ones present in this paper; the obtained results showed the average rolling resistance coefficient to be close to 0.012 for the simplest bearing design. The bearing withstood 2000 1 mm back and forth cycles under constant load before complete breakdown, compared to our bearing, which withstood approximately 9000 cycles without any significant signs of wear; we suppose this decrease in wear is related to the use of cages, but additional measurements should be performed to conclude this. Other differences include a thin metallic layer used instead of bulk metal, as well as different metal used for contact and higher humidity. The authors showed that using extremally thin (below 150 nm) protective films could improve the frictional characteristics of the bearing, but with the increase in coating thickness this effect was lost. The authors also shown that using organic coating on the raceways can improve the frictional behaviour of the system. In article [12], Lin et al. tried to characterize the rolling resistance coefficient in linear ball bearings using stainless steel 285 μm balls together with silicon raceways. The authors obtained a dynamic simplified rolling resistance coefficient equal to 0.045 under 40 g load, which is significantly larger than the values observed during our research; in this case, however, significantly larger rolling elements were used, and a swap of materials between balls and discs should be mentioned. One additional reason for this higher rolling resistance coefficient observed is lower sphericality of micro balls fabricated from stainless steel than those made from soda-lime glass. Another team of researchers studied the performance of borosilicate glass microspheres with approximately 50 μm diameter rotated between the two circular silicon plates (15 mm diameter) under 15, 20, and 30 g loads in high-humidity conditions. They obtained average rolling resistance coefficients equal to 0.008, 0.007, and 0.005, compared to 0.25 for silicon–silicon sliding [13]. The obtained results fall within the ranges observed during this investigation, which can indicate that additional friction introduced by the cage might be compensated for by the additional operational stability introduced by the cage; the mentioned study was, however, carried out for Si discs. Ghalichechian et al. investigated friction in linear and rotary micromotors using 285 μm balls, obtaining a simplified rolling resistance coefficient approximately equal to 0.017 [26].

## 5. Conclusions

We have demonstrated the design and tribological characteristics of the under-1 mm micro ball bearing. The most optimal demonstrated bearings design contained seven glass balls placed between stainless steel plates using a cage with a height equal to 40 μm with no additional coating. It showed an average simplified rolling resistance coefficient $\bar{\mu }=0.002$, with the observations not exceeding $\mu_{0.99}=0.008$ with 99% probability. The created bearing showed good operational stability, and the usage of strong materials ensured low wear even after extended operation. The use of solid lubricant on the separators did not introduce additional performance increases. Similar bearings can be easily replicated and manufactured in large quantities due to good accessibility, the low cost of the materials used, and simple manufacturing methods based mostly on laser cutting. This type of bearing could be easily used in microturbines to improve mechanical performance and to limit wear, or it could be used in micro-robotic joints, such as the one developed as part of the UWIPOM2 project. The proposed device paves the way for future possible research involving operation in highly humid or liquid environments, the impact of the raceways, or much more extended wear and different types of coatings.

## Figures and Tables

**Figure 1 micromachines-14-01775-f001:**
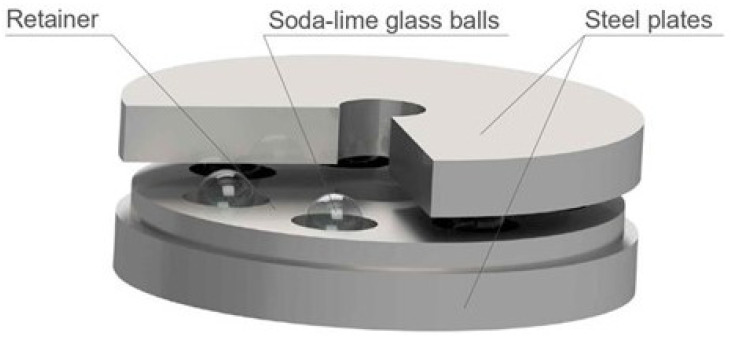
Proposed micro ball bearing design.

**Figure 2 micromachines-14-01775-f002:**
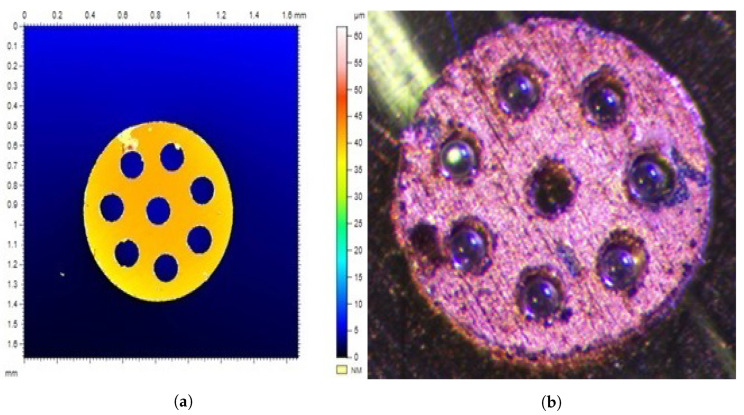
The manufactured cages. (**a**) displays height profile and the dimensions of the manufactured 40 μm cage, while (**b**) shows retainer coated using copper fitted with soda-lime glass balls.

**Figure 3 micromachines-14-01775-f003:**
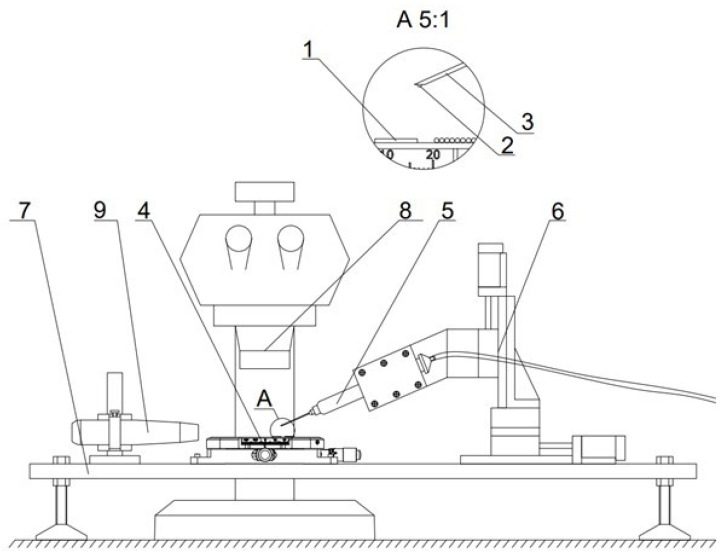
Designed assembly station 1—bearing, 2—ball, 3—needle, 4—precise positioner, 5—syringe, 6—XYZ motion stage, 7—leveled table with rubber dampeners to limit noise, 8—stereoscopical optical microscope with camera, and 9—camera. The device was self-assembled by the authors of the publication.

**Figure 4 micromachines-14-01775-f004:**
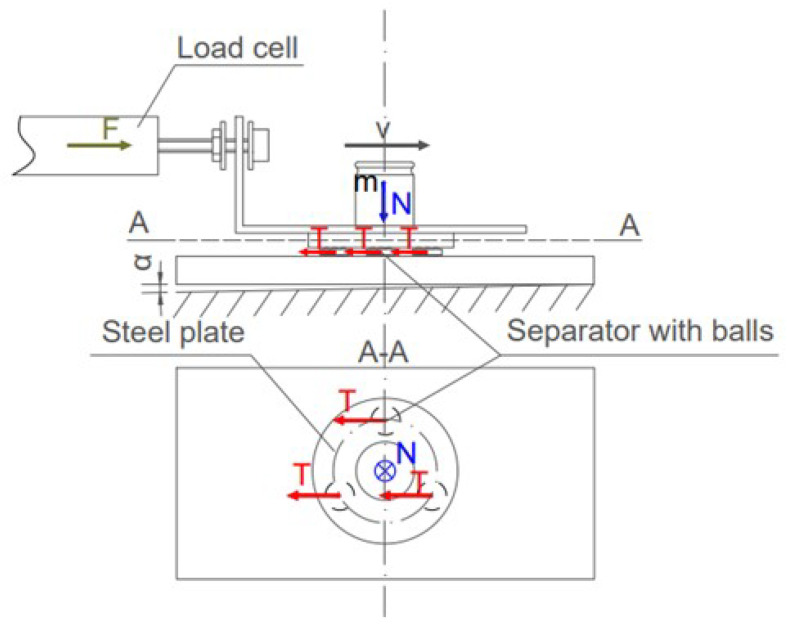
Measurement setup. The bearings are pushed with constant velocity *v* using motorized linear stage. Load cell exerts force *F* equal to the frictional force *T* generated by the bearings loaded with mass *m* exerting normal force *N*. Used measurement method removes misalignment error described as α.

**Figure 5 micromachines-14-01775-f005:**
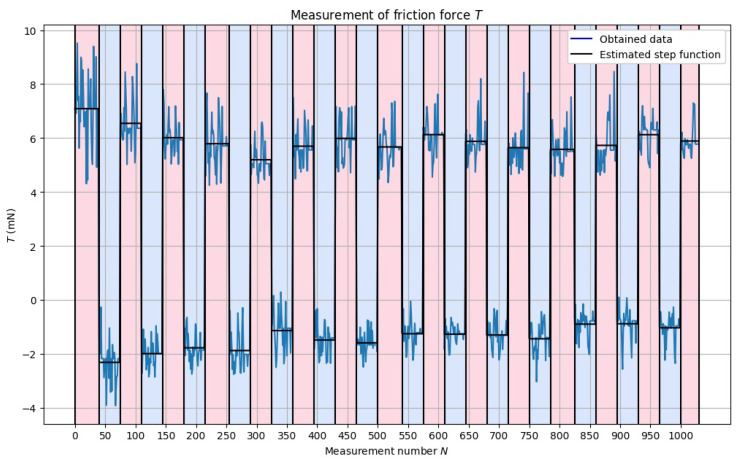
Example of obtained signal for the 30 μm cage under 110 g of load.

**Figure 6 micromachines-14-01775-f006:**
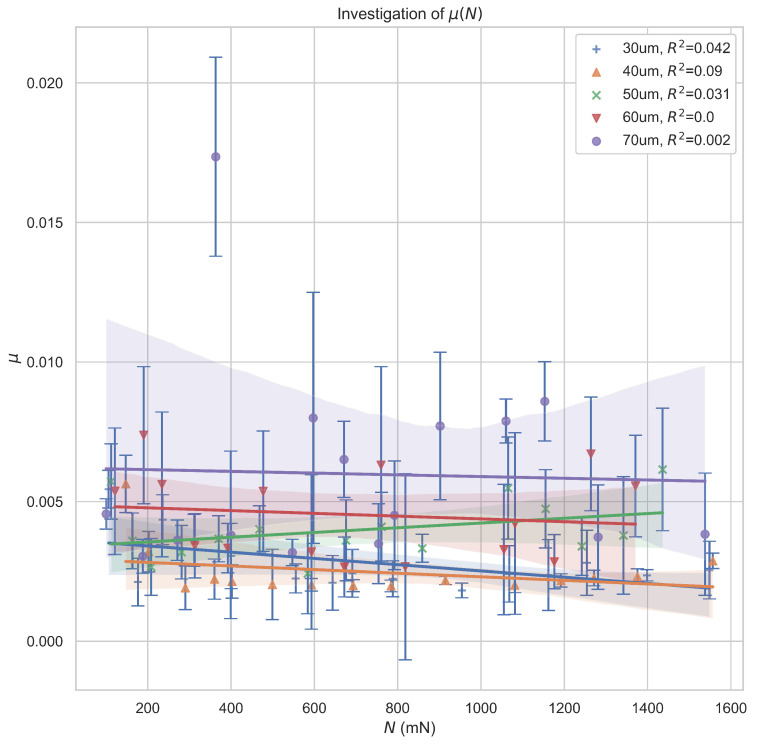
The observed relationship between the simplified rolling resistance coefficient and normal load for different steel cage thickness.

**Figure 7 micromachines-14-01775-f007:**
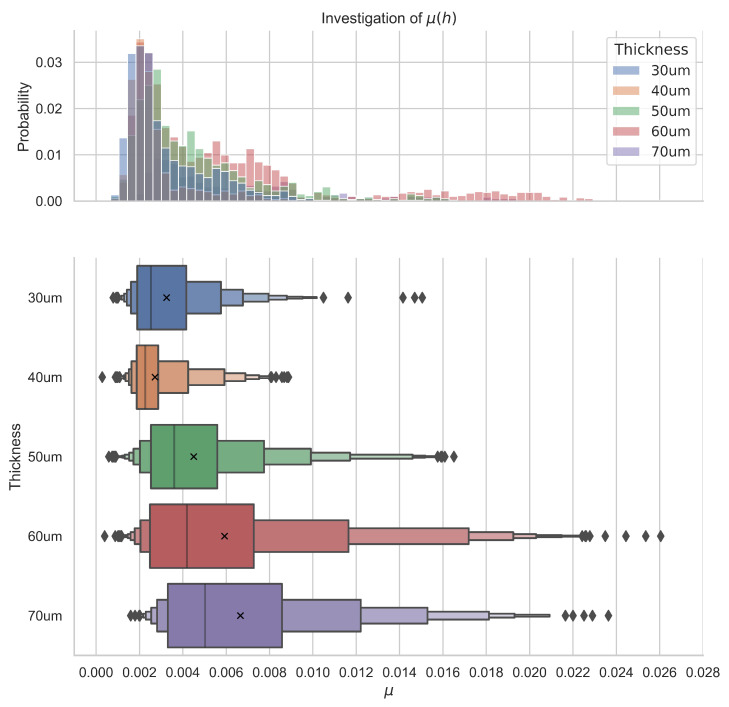
Differences in observed distributions of rolling resistance coefficients as a function cage thickness made from steel. The structure of the used boxplots is explained in [22].

**Figure 8 micromachines-14-01775-f008:**
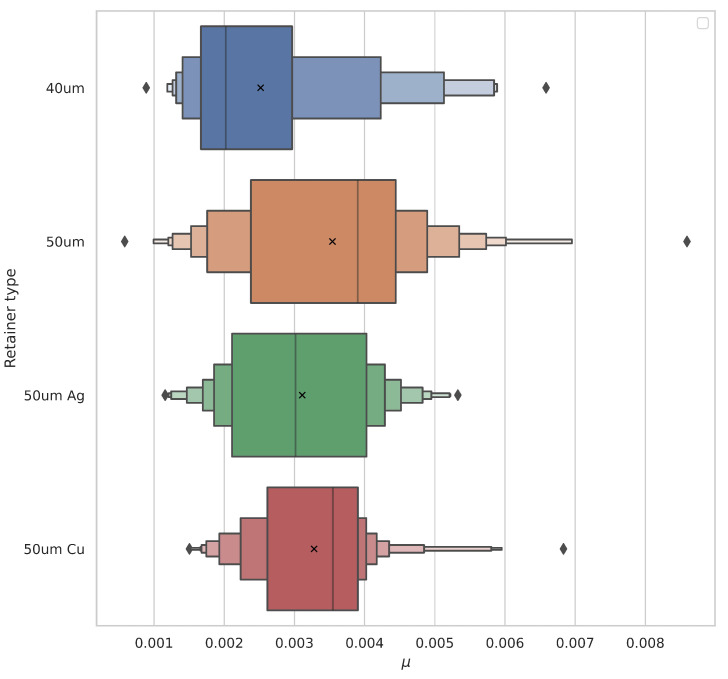
Differences in observed distributions of rolling resistance coefficients as a function coating material. The graph clearly shows that coated 40 μm steel cages indicate mechanical behaviour closer to the cages with 50 μm thickness due to the increase in size.

**Figure 9 micromachines-14-01775-f009:**
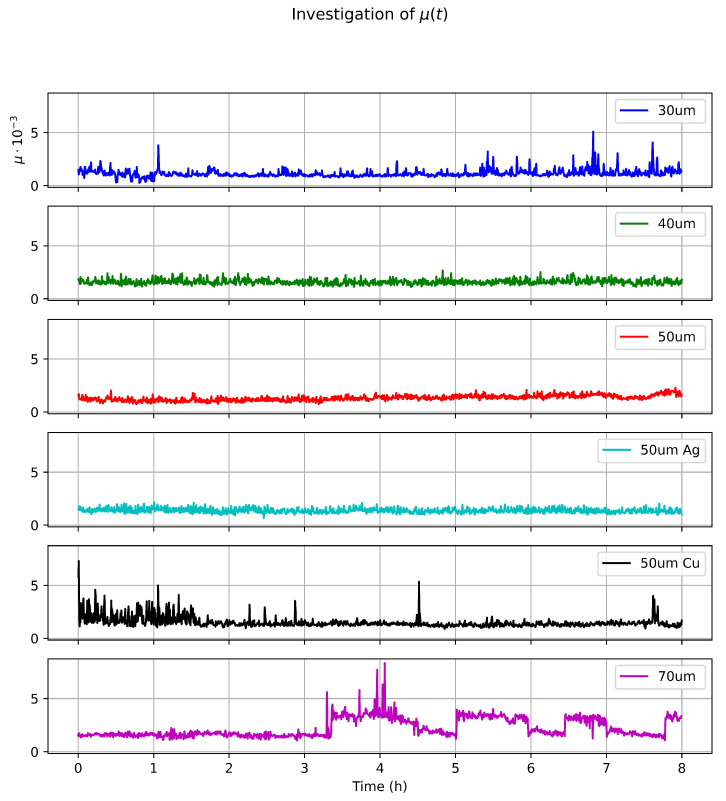
Results of long-term friction measurements.

**Table 1 micromachines-14-01775-t001:** Dimensions of the manufactured elements. The values in brackets indicate uncertainty of the measurement.

Cage Type	$\bar{h}$ (μm)	$\bar{d_{\mathrm{out}}}$ (μm)	$\bar{d_{b}}$(μm)
70 μm	79 (4)	798 (6)	134 (7)
60 μm	66 (4)	801 (6)	138 (7)
50 μm	55 (6)	799 (5)	134 (6)
40 μm	46 (2)	801 (7)	136 (7)
30 μm	35 (2)	801 (7)	131 (7)
40 μm Cu	52 (2)	798 (5)	125 (6)
40 μm Au	55 (4)	797 (6)	124 (5)
Disc	994 (12)	955 (15)	-

## Data Availability

Not applicable.

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
