# Peer review of "Tribological Characterization of Micro Ball Bearings with and without Solid-State Lubrication"

_micromachines, 2023, doi:10.3390/mi14091775_

Round 1

Reviewer 1 Report

The investigation of microball bearings is valuable for the light-mass and micro machines. Some problems should be answered.

1)        What is the operating environment? The coating will be oxidized especially by the action of rubbing heat in air.

2)        In Fig.1, how to find the internal defect of glass ball?

3)        In Fig.7, the data variation is large. Why?

4)        In L237, what’s the meaning of “2000 1 mm cycles”?

The manuscript has some typos. Pls check them carefully.

Author Response

Dear reviewer,

many thanks for your time and for your very pertinent comments.

See our responses and changes acknowledgement on the attached PDF.

Thanks again.

Reviewer 2 Report

This paper mainly investigated the tribological characterization of micro ball bearings with and without solid-state lubrication. However, this work lacks of deep discussion. Thus, I think that this paper should be not published in the present form.

 Comments:

1.More information about the micro ball bearings with and without solid-state lubrication can be given in the introduction section. The authors also should introduce some commonly used solid lubricants.

2.What is the novelty of this paper? The difference from similar studies in the literature should be clearly demonstrated.

3.Please provide the humidity for the test.

4.The author should use characterization techniques to analyze the experimental results carefully.

Author Response

(The authors gave the same response as above.)

Reviewer 3 Report

The paper presents experimental investigations on the friction coefficient on ball micro-bearings. Tribology involves Friction & Wear & Lubrication. At the micro/nano- scale also the surface parameters and adhesion. Only the first aspect was considered in paper as the rolling resistance coefficient. Moreover, based on the AFM, the surface parameters as roughness of the ball and the cages must be provided. No anything about the wear resistance.

Before a possible publication some aspects must be considered for the paper improvement:

1)      In this paper is used different name as: micro ball; microball, micro-ball, micro-scale ball.

2)      Are there no raceways on the bearing rings? How is centered the upper disc (rotating one) in relation with the lower one (the fixed disc)?

3)      The quality of figures must be improved (Fig.2)

4)      Line 77: The optimal process parameters were determined empirically. Based on what?

5)      Line 84: What surface parameters can be observed from Fig.2?

6)      What do the values in brackets in Table 1 represent? More precise explanations should be given.

7)      It is not clear if the assembly device in Figure 4 was made by the authors?

8)      In equation 1, T is the resistance tangential force. But this is the classical expression for friction force. Why don’t the authors use this name of the force? Moreover, expression 1 is the expression for the friction coefficient. What is the difference between the friction coefficient and rolling resistance coefficient?

9)      It is not clear if the bearing is designed for rotational or translational movement? Axial bearings are usually used for rotary motion and ball or roller guides for linear motion. If this bearing is recommended for linear movement between discs, the linear motion is limited and the load on the balls is not uniform. Why is linear movement used for testing (Fig.5)?

10)   Figure 5 - how was the mass centered on the upper disk? In figure 5, was the adhesion force considered during the test?

11)   Figure 6 – What is the Sample number N?

12)   Figure 7 – Investigation of µ(N)? What represents µ(N)? Maybe µ versus N. The same for Fig. 8. Why is such difference at 400mN for 70 µm (Fig.7)?

13)   Line 188 -- We have defined three scenarios one when the height of the cage h is greater than the ball radius r. But in this situation, we have the other kind of friction upper disc-cage-lower disc. Sliding friction and not rolling friction. The same is for last case h approx. equal with r. What approximatively between them means? If h is little bit higher than r, we are in the first case. If it is smaller, we are in the second one.

14)   Line 206 - influence of the solid lubricants on the device performance was investigated. But no information about the lubrication effect of the friction coefficient was provided.

15)   Section 4.4 – line 220: How the bearing performance was estimated? The friction coefficient is not a parameter to estimate the bearing performance. Sometimes the friction coefficient decreases during time operating and the lifetime decreases but based on material fatigue. Fatigue is the main parameter for out-of-use of bearings.

16)   Line 270 – Why the authors present a conclusion on the effect of lubrication on the friction coefficient if it was not presented in the paper. But it is known that lubrication has a considerable effect both on the macro-scale and on the micro-scale.

must be improved

Author Response

(The authors gave the same response as above.)

Round 2

Reviewer 1 Report

The manuscript could be accepted.

Reviewer 2 Report

The authors have revised the manuscript according to the reviewers' comments.